# The Level of Self-Care among Patients with Chronic Heart Failure

**DOI:** 10.3390/healthcare9091179

**Published:** 2021-09-08

**Authors:** Piotr Pobrotyn, Grzegorz Mazur, Marta Kałużna-Oleksy, Bartosz Uchmanowicz, Katarzyna Lomper

**Affiliations:** 1Department information, Clinical University Hospital, 50-556 Wroclaw, Poland; katuchna@tlen.pl; 2Department of Internal Medicine, Occupational Diseases, Hypertension and Clinical Oncology, Wroclaw Medical University, 50-556 Wroclaw, Poland; grzegorz.mazur@umed.wroc.pl; 31st Department of Cardiology, University of Medical Sciences in PoznanLord’s Transfiguration Clinical Hospital in Poznan, 61-848 Poznan, Poland; marta.kaluzna@wp.pl; 4Department of Nervous System Diseases, Faculty of Health Sciences, Wroclaw Medical University, 51-618 Wroclaw, Poland; bartosz.uchmanowicz@umed.wroc.pl; 5Department of Clinical Nursing, Faculty of Health Sciences, Wroclaw Medical University, 51-618 Wroclaw, Poland

**Keywords:** heart failure, self-care, EHFSc-9 self-care behavior scale, lifestyle

## Abstract

Introduction: In a long-term approach to the treatment of heart failure, importance is given to the process of self-care management and behaviors. The number of rehospitalizations and unscheduled medical visits can be reduced by actively engaging patients in the self-care process. Methods: The study included 403 patients with chronic heart failure (mean LVEF 40.53%), hospitalized in the Cardiology Department. Medical record analysis and a self-report questionnaire were used to obtain basic sociodemographic and clinical data. The European Heart Failure Self-care Behavior Scale, revised into a nine-item scale (EHFScBS-9), was used to evaluate self-care behavior. Results: Analysis of the EHFSc-9 self-care behavior scale showed that the mean score was 49.55 out of 100 possible points (SD = 22.07). Univariate analysis revealed that significant (*p* < 0.05) negative predictors of the EHFScB-9 self-care scale included: male sex (b = −5146), hospitalizations in the last year (b = −5488), NYHA class II (b = −11,797) and NYHA IV class (b = −15,196). The multivariate linear regression model showed that a significant (*p* ˂ 0.05) negative predictor of the EHFScB-9 self-care scale was male sex (b = −5.575). Conclusions: Patients with chronic HF achieve near optimal self-care behavior outcomes. A patient prepared to engage with self-care will have fewer rehospitalizations and a better quality of life.

## 1. Introduction

As a result of the increasing incidence of chronic diseases, there is a growing burden on healthcare systems around the world. Therefore, there is an increasing need for ill people to assume responsibility for their own health through involvement in the process of self-care. The essential importance of self-care in the long-term approach to the treatment of heart failure (HF) is emphasized. However, it is important to note that there are differences in healthcare systems, approaches to patient education, and accessibility to health programs in various countries [1]. Self-care in HF involves taking steps to prevent the consequences of the disease in the first place by taking medication, exercising regularly, monitoring symptoms of the disease (especially weight control and the presence of edema), and managing them skillfully when they occur through self-management of medications or contacting the treating physician [2].

Heart failure is currently one of the most commonly diagnosed chronic diseases. HF in developed countries affects 1–2% of the population, while among people over 70, it affects up to 10% [3]. It should be noted that the prevalence of HF in European countries and the United States of America varies and ranges from 1 to 9 per 1000 person-years [4].

The available data also indicates that despite a significant improvement in the patient’s condition during hospitalization, between 60 and 90 days after leaving the hospital, patients who have had an episode of decompensation experience a high mortality rate and frequent rehospitalizations (15% and 30%, respectively) [5].

The Heart Failure Atlas points to an increase in the incidence of HF that is mainly attributed to the phenomenon of an aging population and longer survival times for patients receiving guideline-directed medical therapy [6]. With advances in treatment modalities and the implementation of Evidence Based Medicine (EBM) into daily clinical practice, it has become possible to increase survival and reduce the hospitalization of patients, but data from the ESC-HF pilot study indicate that the 12-month hospitalization rate remains high (43.9% and 31.9% among hospitalized patients with acute HF and chronic HF) [7]. HF patients are at an increased risk of hospitalization and death, as well as a reduced quality of life. An additional problem is that a significant proportion of patients are elderly, diagnosed with frailty syndrome and coexisting comorbidities [8].

HF is the most common cause of hospitalizations of patients over the age of 65. Contributing factors may include lack of or inadequate organization of ambulatory care. These factors affect the complexity of care and may affect self-care behaviors. As emphasized in “A curriculum for heart failure nurses”, the need to optimize the care of patients diagnosed with HF is one of the biggest challenges in modern cardiology, and patient supervision, including education focused on self-care, will be reflected in improved adherence and rehospitalization rates, among other things [8,9].

The importance and role of self-care in HF was also highlighted in the ESC guidelines. Self-care in HF should focus on adherence to treatment recommendations, lifestyle modification, monitoring of disease symptoms and the ability to respond to HF exacerbations [3]. The number of rehospitalizations and unplanned medical appointments among HF patients can be reduced by actively involving patients in self-care. It has been shown that patients diagnosed with HF who are characterized as more effective in their self-care behaviors maintain a better quality of life, lower mortality and fewer rehospitalizations due to exacerbation of the disease [10,11]. Despite the existing evidence showing the benefits of self-care in heart failure, patients still experience difficulties in observing and performing it, therefore it is insufficient [12]. The evidence also points to difficulties in self-care, particularly in recognizing and responding to disease symptoms that occur, among older people compared to younger HF patients [13].

Therefore, educational programs tailored to the individual patient’s needs are necessary. The provision of accurate knowledge about HF and disease management should be part of the treatment of patients with HF. The knowledge about factors determining the level of patient’s involvement in the treatment process and the level of self-care may be useful in planning health education. There are many studies assessing the level of self-care in heart failure, however only few of them focus on the influence of sociodemographic factors [14]. In the light of the data presented, an attempt was made to assess the level of self-care in patients diagnosed with HF and its determinants.

## 2. Materials and Methods

### 2.1. Study Settings and Participants

The study was conducted in the Department of Cardiology of the University Clinical Hospital in Wroclaw. The criteria for inclusion in the study were: diagnosis of chronic HF during a minimum period of 6 months before the study, aged 18 years and above, informed consent to participate in the study and the patient’s condition not requiring intensive cardiac supervision.

### 2.2. Ethical Consideration

The study was approved by the independent Bioethics Committee (KB 388/2017) of the Wroclaw Medical University, Poland. All participants were informed about the purpose of the study, conduct, and the possibility of withdrawal at any stage. The study was conducted in accordance with the requirements of the Declaration of Helsinki.

### 2.3. Research Tools

The study used a Polish adaptation of the European Heart Failure Self-Care Behavior Scale (EHFScB-9) questionnaire to assess HF patients’ self-care. The “classic” total score on this questionnaire is a number ranging from 9 to 45, with high scores indicating a low level of self-care. For EHFScB-9, there are no standards regarding how many points can be treated as a high score, and how many as an average. The authors of the tool proposed a simple transformation that changes the scale of the result from 9–45 to 0–100 and reverses it. They called this result “inverted standardized”. In this version, the midpoint is 50, and high scores indicate a high level of self-care [15,16,17].

Socio-demographic data (age, sex, education level, place of residence, marital status) and clinical data (duration of disease, medications taken, New York Heart Association (NYHA), functional class, and the results of additional examinations, including left ventricular ejection fraction (LVEF), were collected using a questionnaire and analysis of medical records.

### 2.4. Statistical Analysis

The analysis of quantitative variables (i.e., expressed by number) was performed by enumerating the mean, standard deviation, median, quartile, minimum and maximum. The analysis of qualitative variables (i.e., not expressed by number) was carried out by enumerating the number and percentage of the instances of each value. One- and multi-factor analysis of the effects of multiple variables on the quantitative variable was performed using the linear regression method. The results are presented in the form of parameter values of the regression model with a 95% confidence interval [18].

## 3. Results

### 3.1. Sociodemographic and Clinical Analysis 

The study included 403 patients diagnosed with chronic heart failure (230 men and 173 women) with an average LVEF ejection fraction of 40.53 ± 11.11% and those mainly in II (28.54%) and NYHA Class III (45.91%). The largest group were patients aged 65–70 years (31.27%) and 77–82 years (19.35%) and people living in cities (64.76%). The vast majority were patients who had HF for 1 to 5 years (43.18%) and those with vocational education (34.24%) and high-school education (32.51%). Based on the analysis, the study group was hospitalized for exacerbation (73.45%) and the most commonly taken drugs were beta-blockers (87.34%) and diuretics (83.13%). The basic sociodemographic and clinical data are presented in Table 1.

### 3.2. EHFSc-9 Self-Care Questionnaire Analysis

Analysis of EHFSc-9 self-care questionnaire results showed that the average score was 49.55 points out of 100 possible (SD = 22.07), and the scores ranged from 0 to 100 points. The data is presented in Table 2.

### 3.3. One-Factor Analyses of Self-Care Determinants (EHFScB-9)

Based on linear regression models significant (*p* < 0.05) negative predictors of the EHFScB-9 self-care questionnaire score were male sex (b = −5.146), hospitalizations in the last year (b = −5.488), NYHA class II (b = −11.797) and NYHA class IV (b = −15.196). Significant positive predictors of the EHFScB-9 self-care questionnaire score were disease duration of more than 10 years (b = 8.529), a permanent relationship (b = 6.515), and an increase in LFEV% (b = 0.353). The data are presented in Table 3.

### 3.4. Multifactorial Analyses of Self-Care Determinants (EHFScB-9)

A Multivariate linear regression model showed that a significant (*p* < 0.05) negative predictor of the EHFScB-9 self-care questionnaire score was male sex (b = −5.575). A permanent relationship (b = 5.933), disease duration of more than 10 years (b = 8.918) and an increase in LVEF% (b = 0.359) were significant (*p* < 0.05), positive predictors of the self-care questionnaire score. The data are presented in Table 3.

## 4. Discussion

With the improvement in the effectiveness and availability of heart failure treatment, there has also been a reduction in hospitalizations due to exacerbation of the disease in recent decades. [3] However, heart failure continues to be a global public health problem. Thanks to more effective methods of treatment, the survival of patients has increased, but the course of the disease itself is characterized by frequent exacerbations [19]. Symptoms of increased fluid retention are identified as one of the main causes of rehospitalization in this group, but the importance of concomitant diseases is also highlighted [20].

In addition, patients who demonstrate high levels of self-care also have lower mortality rates and lower rates of re-hospitalizations for disease exacerbations compared to patients with low levels of self-care [21]. It should be noted that for patients with heart failure, self-care may need to be modified depending on the severity of the disease and the treatment used [22].

Despite a number of benefits of implementing the self-care process into the therapeutic plan, many HF patients have trouble following its recommendations [23]. Jaarasma et al., in a meta-analysis of self-care behaviors evaluated using the Self-care of Heart Failure Index or the European Heart Failure Self-care Behavior Scale, reported sub-optimal levels of self-care among 5964 patients diagnosed with HF. The majority of patients surveyed declared that they were taking the medicines as directed, although behaviors related to physical activity and weight monitoring, among other things, were unsatisfactory [2]. On the basis of an analysis of the EHFScBS-9 self-care questionnaire in the group of 270 patients with HF, Uchmanowicz et al. recorded a high average score of 50.39 points. Interestingly, the authors observed a significant relationship between the outcome of the self-care questionnaire and cognitive function—the decrease in self-care levels was accompanied by a decrease in cognitive function [24]. In comparison, our own study achieved a similar average score of the EHFSc-9 self-care questionnaire results. The analysis showed that the mean score was 49.55 out of a possible 100 points. The values obtained in our own study can be attributed to the old age of the study group. In the study, the vast majority were patients over 65 years of age. In the literature, old age is identified as one of the factors causing difficulties in self-care (especially symptom recognition) among patients diagnosed with HF, although it was not a significant determinant of self-care in our study. [13]. The lack of significance of this factor on the self-care process may be due to the fact that a significant proportion of the study group were of high-school grade (32.51%), or higher education (16.38%) and also in relationship (52.11%).

Gender differences in maintaining optimal levels of self-care in a group of patients with chronic heart failure have been previously demonstrated [25]. In our study, in the multivariate analysis, the male sex was shown as a negative predictor of the level of self-care. The available literature indicates controversy in this regard. In the study by Mei et al., men also showed lower self-care scores compared to the female group [26]. However, Lee et al. documented that the male sex was better at self-care maintenance [27]. Self-care among patients with chronic HF is an important component for optimal results of therapy. Sociodemographic factors influencing adherence to the self-care process also include social support. The type and severity of disease symptoms in HF make it imperative that the self-care process be accomplished with the support of life partners, among others. It has been documented that those patients who receive social support are better at maintaining self-care [11,28]. In the Mei et al. study already cited, social support was a positive factor in maintaining self-care levels [26]. Similarly, in the Gallagher et al. study, patients with high levels of social support showed better self-care than patients with low or moderate support. Interestingly, among patients with a life partner, only 49% declared a high level of support [29]. Based on a self-study, the effects of social support (having a partner) on self-care levels were also observed. Being in a relationship was an important positive determinant of the result of the EHFSc-9 questionnaire in the multi-factor analysis. It is important to emphasize that the presence of social support is not in itself sufficient to influence self-care among patients with HF. Moreover, the involvement of caregivers, especially life partners, should constitute an integral part of the care and treatment of HF [29].

The European Society of Cardiology Guidelines on Heart Failure recommend a number of behaviors of a self-care nature that may be crucial in achieving satisfactory HF treatment outcomes. Among other things, the role of smoking, the use of nutritional recommendations, and the adjustment of the dosage of diuretic drugs are indicated, but attention is also paid to the ability to regularly monitor conduction symptoms or daily weight measurements [3]. Being able to recognize and respond to disease symptoms appears to be central to self-care. Thus, high scores in self-care should also be expected in patients who have experienced a long duration of the disease. A study conducted among 116 outpatient HF patients using the Self-Care of Heart Failure Index questionnaire found higher self-care values in patients with a longer duration of the disease [30]. The results of our own study were in the line with the available literature—in the multifactorial analysis, the disease duration of over 10 years was a significant determinant of higher values of self-care.

In summary, male gender was a negative predictor of self-care. This phenomenon can be explained by the fact that men are more likely than women to be economically active, work longer, and therefore spend less time on health education. Therefore, they spend less time educating themselves about disease. Among positive determinants of self-care, we identified relationship status, longer disease duration (more than 10 years), and increased LVEF. Social support itself and the support of a family member are important factors relevant to the self-care process. Additionally, long disease duration may be associated with a longer time to come to terms with the disease and acceptance of the disease, and these are determinants of self-care. An increase in LVEF indicates that patients are adhering to treatment recommendations, taking medications, and regularly participating in their activities.

## 5. Conclusions

In our study, patients with chronic HF achieve almost a “midpoint” of self-care. Recognition of the factors determining the level of patient involvement in the treatment process and the level of self-care may be useful in planning health education. A patient prepared for self-care will experience fewer rehospitalizations and a better quality of life.

We have also added additional informations in Appendix A.

## Figures and Tables

**Table 1 healthcare-09-01179-t001:** Sociodemographic and clinical data of the study group (n = 403).

Parameter		Total (N = 403)
LVEF [%]	SD	40.53 ± 11.11
	Median	40
	Quartile	32–48
Sex	Women	173 (42.93%)
	Men	230 (57.07%)
Age	60–64 years	57 (14.14%)
	65–70 years	126 (31.27%)
	71–76 years	62 (15.38%)
	77–82 years	78 (19.35%)
	83–88 years	63 (15.63%)
	Over 88 years	17 (4.22%)
Marital status	Single	193 (47.89%)
	In a relationship	210 (52.11%)
Place of residence	Village	142 (35.24%)
	Town	261 (64.76%)
Education	Primary	67 (16.63%)
	Vocational	138 (34.24%)
	High-school	131 (32.51%)
	Higher	66 (16.38%)
	No data	1 (0.25%)
Duration of disease [years]	Less than a year	47 (11.66%)
	1–5 years	174 (43.18%)
	6–10 years	82 (20.35%)
	Over 10 years	100 (24.81%)
Hospitalizations in the last year	No	107 (26.55%)
	Yes	296 (73.45%)
Medications taken	Beta-blockers	352 (87.34%)
	ACE-I, ARB	245 (60.79%)
	Digoxin	71 (17.62%)
	Diuretics	335 (83.13%)
NYHA Class	I	17 (4.22%)
	Ⅱ	115 (28.54%)
	III	185 (45.91%)
	IV	85 (21.09%)
	No data available	1 (0.25%)

Abbreviations: LVEF: left ventricle ejection fraction; NYHA Class: New York Heart Association Class.

**Table 2 healthcare-09-01179-t002:** EHFSc-9 Self-Care Questionnaire Results.

EHFSc-9 [Points]								
N	Data Deficiencies	Average	SD	Median	Min	Max	Q1	Q3
403	0	49.55	22.07	47.22	0	100	33.33	63.89

**Table 3 healthcare-09-01179-t003:** Results of the single-factor analysis and multifactor analysis of EHFScB-9 self-care questionnaire determinants in the study group (n = 403).

Parameter		Single-Factor Models				Multifactor Model			
		Parameter	95% CI		Q	Parameter	95% CI		*p*
Sex	Women	ref.				ref.			
	Men	−5.146	−9.475	−0.816	0.02 *	−5.575	−10.359	−0.791	0.023 *
Age	Under 65 years	ref.				ref.			
	65–70 years	3.38	−3.496	10.256	0.336	3.45	−3.46	10.36	0.328
	71–76 years	3.254	−4.651	11.159	0.42	0.333	−7.809	8.474	0.936
	77–82 years	1.996	−5.51	9.503	0.603	1.367	−6.477	9.21	0.733
	83–88 years	−3.697	−11.572	4.178	0.358	−4.928	−13.215	3.359	0.245
	Over 88 years	−8.342	−20.247	3.563	0.17	−7.414	−19.863	5.035	0.244
Marital status	Single	ref.				ref.			
	In a relationship	6.515	2.243	10.786	0.003 *	5.933	1.348	10.518	0.012 *
Place of residence	Village	ref.				ref.			
	Town	−0.631	−5.147	3.885	0.784	0.857	−3.807	5.522	0.719
Education	Primary	ref.				ref.			
	Vocational	−5.463	−11.89	0.965	0.097	−4.492	−10.992	2.008	0.176
	High-school	−1.404	−7.887	5.079	0.671	−2.106	−8.645	4.434	0.528
	Higher	1,456	−6.03	8.942	0.703	2.783	−4.991	10.556	0.483
Duration of the disease [years]	Less than a year	ref.				ref.			
	1–5 years	7.092	0.005	14.179	0.051	4.227	−2.909	11.364	0.246
	6–10 years	4.299	−3.588	12.187	0.286	5.23	−2.732	13.192	0.199
	Over 10 years	8.529	0.905	16.153	0.029 *	8.918	1.352	16.485	0.021 *
Hospitalizations in the last year	No	ref.				ref.			
	Yes	−5.488	−10.344	−0.632	0.027 *	−4.197	−9.288	0.894	0.107
Medications: Beta blockers	No	ref.				ref.			
	Yes	3.104	−3.378	9.586	0.349	5.061	−1.384	11.507	0.125
Medications: ACE-I, ARB	No	ref.				ref.			
	Yes	−1.402	−5.819	3.015	0.534	−0.958	−5.34	3.425	0.669
Medications: Digoxin	No	ref.				ref.			
	Yes	3.061	−2.594	8.716	0.289	2.843	−3.019	8.705	0.342
Medications: Diuretics	No	ref.				ref.			
	Yes	−0.097	−5.857	5.664	0.974	2.429	−3.681	8.54	0.436
LVEF	[%]	0.353	0.162	0.545	< 0.001 *	0.359	0.137	0.582	0.002 *
NYHA Class	I	ref.				ref.			
	II	−11.797	−22.937	−0.658	0.039 *	−7.47	−18.763	3.823	0.196
	III	−8.524	−19.389	2.341	0.125	−2.262	−13.722	9.198	0.699
	IV	−15.196	−26.586	−3.806	0.009 *	−5.493	−17.744	6.759	0.38

* Statistically significant correlation (*p* < 0.05); Abbreviations: LVEF: left ventricle ejection fraction; NYHA Class: New York Heart Association Class; ACE-I: angiotensin-converting-enzyme inhibitors; ARB: angiotensin receptor blockers.

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
