# Peer review of "The Level of Self-Care among Patients with Chronic Heart Failure"

_healthcare, 2021, doi:10.3390/healthcare9091179_

Round 1

Reviewer 1 Report

This is a manuscript that report the level of self-care of heart failure patients. This group's earlier report in 2014 concluded that "self-care–related behaviors may be optimized as a result of appropriate educational activities", and as there are significant contribution of male sex, being in a  permanent relationship,  disease duration of more than 10 years  were significant and positive predictors of the self-care questionnaire score, one may wonder whether the above predictors were also correlated with educational activities, e.g. longer disease duration means more changes for educational activities, and male sex (perhaps more active as a breadwinner, might prevent them from attending educational activities) and being in a permanent relationship might mean better reminding of attending educational activities by long-term partners. Thus it might be important to check how independent the newly reported preditors are in relation to the earlier report of educational activities as a factor. 

One thing not very clear is that this report says mean score was 49.55, while the 2017 reported "The respondents’ mean score in the self-care scale was 50.39 points ". The author mentioned the two numbers and said "considered as a high level of self-care, although still at a low level", without saying it is comparable to the previous report (i.e. consistent or not), thus making it a bit confusing in the discussion section. 

Under "what's is new here" the author listed "implement a sick self-care process". It might be better to say "heart failure self-care process" or perhaps a "healthy heart self-care process"; because "sick self-care process" has a certain negative connotation to it. 

Also under "What's is new here" the author listed "better patient education" as a solution; however, the paper does not really measure anything about patient education but rather it measures self-care process (or lack of self-care) and this might lead to better awareness of the patients for improving their self-care process. Therefore I think rewording this part might be better.

Reviewer 2 Report

I have read with interest the manuscript "The level of self-care among patients with chronic heart failure.". This is an interesting topic that can make a potential contribution to clinical practice. However, I have some concerns about this work.

1.

In the introduction, I couldn't understand the hypothesis of this study. It is explained that self-care affects readmission, mortality and quality of life. However, previous studies did not explain whether self-care was associated with other parameters (gender, duration of illness, etc.). Can you add and explain references related to self-care?

2.

Is it possible to add the BNP value for Research tools? BNP indicates the severity of heart failure.

3.

In results factors obtained from multivariate analysis were sex, Marital status, Duration of the Disease, and LVEF. These are considered to be factors that are difficult to change. We need to state how to utilize it in discussion. Can you explain these?

4.

In 4. Discussion, this study did not extract age as a factor in self-care, why? Can I add a description?

5.

What is the novelty of this study? Will the factors associated with self-care in patients with heart failure be novel? In the discussion, can the discussion clarify the difference from previous study ?

Round 2

Reviewer 2 Report

I think the content of this Article has improved.
